# People Flow Trend Estimation Approach and Quantitative Explanation Based on the Scene Level Deep Learning of Street View Images

Chenbo Zhao [1,*], Yoshiki Ogawa [2], Shenglong Chen [1], Takuya Oki [3] and Yoshihide Sekimoto [2]

[1]  Department of Civil Engineering, The University of Tokyo, Tokyo 153-8505, Japan
[2]  Center for Spatial Information Science (CSIS), University of Tokyo, Tokyo 153-8505, Japan
[3]  School of Environment and Society, Tokyo Institute of Technology, Tokyo 152-8550, Japan
*   Correspondence: cbzhao@iis.u-tokyo.ac.jp

**Abstract:** People flow trend estimation is crucial to traffic and urban safety planning and management. However, owing to privacy concerns, the collection of individual location data for people flow statistical analysis is difficult; thus, an alternative approach is urgently needed. Furthermore, the trend in people flow is reflected in streetscape factors, yet the relationship between them remains unclear in the existing literature. To address this, we propose an end-to-end deep-learning approach that combines street view images and human subjective score of each street view. For a more detailed people flow study, estimation and analysis were implemented using different time and movement patterns. Consequently, we achieved a 78% accuracy on the test set. We also implemented the gradient-weighted class activation mapping deep learning visualization and L1 based statistical methods and proposed a quantitative analysis approach to understand the land scape elements and subjective feeling of street view and to identify the effective elements for the people flow estimation based on a gradient impact method. In summary, this study provides a novel end-to-end people flow trend estimation approach and sheds light on the relationship between streetscape, human subjective feeling, and people flow trend, thereby making an important contribution to the evaluation of existing urban development.

**Keywords:** deep learning; deep learning explanation; people flow estimation; street view images; streetscape impact analysis

## 1. Introduction

Rapid urbanization in recent years has highlighted the importance of people flow trend analysis, which can provide crucial information for urban planning [1], disaster management [2], and traffic planning [3]. Additionally, since 2019, people flow studies have played important roles in COVID-19 pandemic estimation and management [4–6]. Thus, the development of an efficient automatic people flow estimation approach and generalization is globally necessary.

However, the direct counting of people flow trend data is difficult. This is because spatiotemporal people flow trend data are primarily generated from global position system (GPS) [7], mobile networks [8], and person trip data [9], which inevitably involve private user information, such as location, movement patterns, and smart phone cookies. To mitigate these issues, people flow estimation methods have been developed using other approaches, such as by cellular probe data [10], probabilistic models [11], and multi-agent simulators [12]. However, these estimation methods are also limited by issues such as privacy protection, the requirement of additional devices to receive their input data, and the complexity of the estimation processing. For example, the high cost of cellular probe over the entire study area and the difficulty in reproducing the non-end-to-end model

in a new area because of the complexity of the intermediate parameter fine-tuning are daunting challenges.

Therefore, a more efficient end-to-end people flow estimation approach is required that does not rely on personal privacy data. Previous studies have combined people flow with street view images [13,14]. These deep learning methods primarily concentrate on the people being monitored or those in street view images, and use object detection and tracking algorithms to count the number of people appearing in the images. However, these methods only consider surface information, such as the number of people in the images, and do not delve into the deeper semantic information, including: (i) the number of people who work or stay in the buildings or pass the roads, (ii) the type of streetscape people that tend to live, work, or pass by, and (iii) the subjective feeling of people in different people flow situations.

In this study, we developed a high semantic level end-to-end people flow estimation approach based on street view images. Subsequently, we validated this approach over a wide area containing over 1.5 million images. Our estimation does not count the number of people appearing in the picture; rather, we obtained a comprehensive people flow trend of a $250 \times 250$ m$^2$ area. It accounts for the people working in the buildings or passing the roads in these areas. To further verify the robustness of our approach, we reprocessed the source data according to four different time and movement patterns based on day/night time and stay/move movement patterns, and reproduced the training and inferencing respectively. To improve the performance of the estimation model, we used a Siamese- based algorithm to extract the subjective feeling scores from street view images and incorporate these data into our deep learning models. Furthermore, to analyze the deep learning processing and results, we implemented semantic segmentation on the images. The semantic segmentation and subjective feeling score results are included as ancillary data. Combining these data with the gradient-weighted class activation mapping (Grad- CAM) [15] deep learning visualization method, we developed a quantitative deep learning explanation approach to explain the three problems discussed previously that are not considered by existing methods. In addition, we compared our approach with a statistically reasonable L1-based sparse model to verify the rationality of the explanation approach. Connecting the street view images, people flow trend, and the pipeline of the deep learning explanation helped provide important information for related to street view and people flow study that can be generalized to other fields.

The primary contributions of this study are as follows:

- We developed an efficient scene-level end-to-end deep learning people flow trend estimation approach based on street view images and human subjective scores.
- We developed a subjective feeling score extraction method and implemented semantic segmentation to provide ancillary information for the model.
- We implemented Grad-CAM and combined it with the semantic segmentation and subjective score extraction results; we also developed a novel quantitative deep learning explanation approach to explain the discussed models and used L1-based sparse modeling to verify rationality.

The remainder of this paper is organized as follows: Section 2 provides an overview of related studies; Section 3 describes the dataset used in this study; Section 4 describes the methodology; Section 5 describes the experiments and results. Section 6 discusses the results and Section 7 presents our findings.

## 2. Related Work

Previous methods on people flow trend estimation primarily focus on people flow statistics based on population grid unit data from census or counting [16]. Through these statistical methods, some studies used areal weighting to construct population grid data [17,18], whereas others used choropleth mapping methods to generate population density maps [19,20]. Those based on choropleth mapping used muti-source ancillary data such as remote sensing images, land cover, urban extent, and accessibility data to

redistribute population counts within the census units. Furthermore, considering these population statistics methods and ancillary data, some studies developed random forest estimation methods to perform higher resolution population mapping [21]. Although the people flow trend-estimation results can be generated by these statistics and high-resolution population maps [22], they exhibit a few limitations: (i) the methods in these studies are not real time; census and ancillary data usually lag behind the time of interest significantly; (ii) ancillary data generation requires ancillary devices, such as satellites and drones, whose high cost renders the reproduction of these methods difficult in developing countries.

In recent years, advances in sensing and analysis technologies have facilitated the development of devices for collecting multi-source related data and intelligent methods to estimate people flow trend. Such location-based big data with respect to people can be generated owing to the increased accessibility of location-aware mobile devices [23]. High-tempora- resolution people flow maps can be generated by reconstructing trajectories from mobile phone records [24]. Yao et al. [25] used a random forest algorithm to analyze the Baidu points-of-interest (POIs) and real time Tencent user density in order to downscale the street-level people flow trend to the grid level. Multi-source data and intelligent analysis methods enable faster people flow trend estimation with higher spatiotemporal resolution. However, the following limitations exist: (i) many of these estimation methods require complex input data or data preprocessing, rendering fine-tuning and generalization difficult; (ii) relevant laws on privacy protection may render certain individual datasets unavailable. Thus, it is necessary to develop generalized end-to-end approach that does not rely on private data. Thus, the combination of using street view images and deep learning methods can potentially prove to be an attractive solution.

Deep learning has far outperformed traditional algorithms in the field of computer vision (CV). Considering image classification as an example, the ImageNet large scale visual recognition challenge (ILSVRC) conducted in 2010, which contained a 1.4 million natural image dataset labeled across 1000 classes, is the benchmark competition in the field of image classification. In this competition, the application of CNN greatly improved the score of AlexNet in ILSVRC [26], after which CNN algorithms have been widely used in the field of CV.

CNN is a trainable feedforward neural network with a deep structure and convolution computation, which is a representative algorithm of deep learning [27]. The CNN feature extraction stage comprises at least three types of layers: convolution, pooling, and non-linear activation function. Further, in image classification tasks such as in the current study, fully connected (FC) layers function as multilayer perceptrons (MLPs) connected with feature extraction stages to complete the final classification task. Another type of deep learning algorithm, ViT [28], proposed in 2020, works effectively in many CV tasks and may become better than CNN in the ImageNet Large Scale Visual Recognition Challenge (ILSVRC). ViT reached the state-of-the-art, with up to 91.00% accuracy [29].

On the other hand, as the deep learning methods are typically used end-to-end algorithms, many direct classification and regression tasks were implemented through them. Considering the regression task as an example, object detection in computer vision is a type of regression where the inputs are only images. Through object detection algorithms, bounding boxes of interested objects can be directly understood [30,31]. Based on the object detection algorithms, many studies proposed some deep learning based people flow counting methods. Hara et al. applied convolutional neural networks (CNNs) and long-short-term-memory (LSTM) to implement sidewalk-level people flow estimation. Zhang et al. [32] proposed a multi-scale convolution kernel DCNN to capture different receptive fields to finally obtain the dense people flow estimation results. However, these methods only consider the surface content in the images, such as tracking the number of people or vehicles appearing in these images, while omitting the people inside these vehicles or buildings. Moreover, as a deep learning algorithm is similar to a black box with respect to the internal aspects of training and inference processing, these deep learning-based methods are usually used to perform certain minor modification on the computer vision

baseline algorithms, declaring that they obtain an effective performance on their own dataset, without explaining the internal process.

Accounting for this issue in deep learning algorithms, if the training and inferencing can be visualized, the output results can be further understood. There are three primary approaches in this regard: convolution visualization, feature map visualization, and class activation map. Wang et al. [33] developed a CNN explainer interactive system to visualize the convolution and CNN architectures. Zeiler et al. [34] mapped the feature maps back to the input pixel space, exposing the aspect of the input pattern that originally caused a certain activation in the feature maps. However, studies that aim to provide a quantitative explanation of deep learning models are still lacking in both computer vision and people flow estimation field.

To explain the deep learning model black box in the people flow estimation study, it is promising to connect the human subjective perception of the images to the model, thus deeper understanding of the model can be obtained. In many studies, subjective scores significantly contribute toward the analysis of landscape [35], attractiveness [36], and housing price [37], which shows the prospect of subjective scores. In this study, we extracted subjective scores to facilitate the improvement of the proposed people flow estimation model and deep learning explanation.

Human subjective scores represent quantitative emotive responses to street view images. However, emotive responses are difficult to quantify. To address this problem, Dubey et al. [38] quantified the emotive response to images by ranking them. They proposed a Siamese CNN to convert the quantification task into a classification task. The output of this Siamese CNN is a logit value that indicates the emotive response comparation between the two input images, that is, whether Image 1 looks safer, livelier, more boring, wealthier, more depressing, or more beautiful than Image 2. Inspired by this mechanism, human subjective score extraction model can be incorporated into the people flow study.

Based on above discussion, we developed a novel method to estimate the people flow with a high semantic level that also explains the deep learning processing and results.

## 3. Data

This study was deployed in Kōchi, Japan (133.39°E 33.46°N to 133.63°E 33.68°N, WGS84) in an area of 25.731 × 29.604 = 761.74 km$^2$.

### 3.1. People Flow Trend Data

The people flow data were extracted from the 2012 mobile phone Global Positioning System (GPS) logs called "Konzatsu Analysis (R)" provided by ZENRIN DataCom Co., LTD.

"Konzatsu-Tokei (R)" Data refers to people flows data collected by individual location data sent from mobile phone under users' consent, through Applications provided by NTT DOCOMO, INC. Those data are processed collectively and statistically in order to conceal the private information.

Original location data are GPS data (latitude, longitude) sent in about every a minimum period of 5 min and does not include the information to specify individuals.

※ Some applications such as "docomo map navi" service (map navi & local guide).

### 3.2. Street View Image Data

This study used 1,523,882 street view images of Kōchi, Japan, obtained from the Zenrin Corporation (2012). The Zenrin Corporation provides a comprehensive collection of street view images throughout Japan. The images were captured at 2.5 m intervals using a 360° camera mounted on a vehicle roof as it drove along the streets. Each image was annotated and geotagged with textual information such as latitude and longitude, measured using the GPS, vehicle azimuth, and shooting time. The original images were panoramic in the jpeg tar format with 2700 (height) × 5400 (width) pixels.

## 4. Methodology

The workflow of our study can be divided into three parts: people flow trend estimation, ancillary data generation and model improvement, and deep learning processing and results explanation (Figure 1).

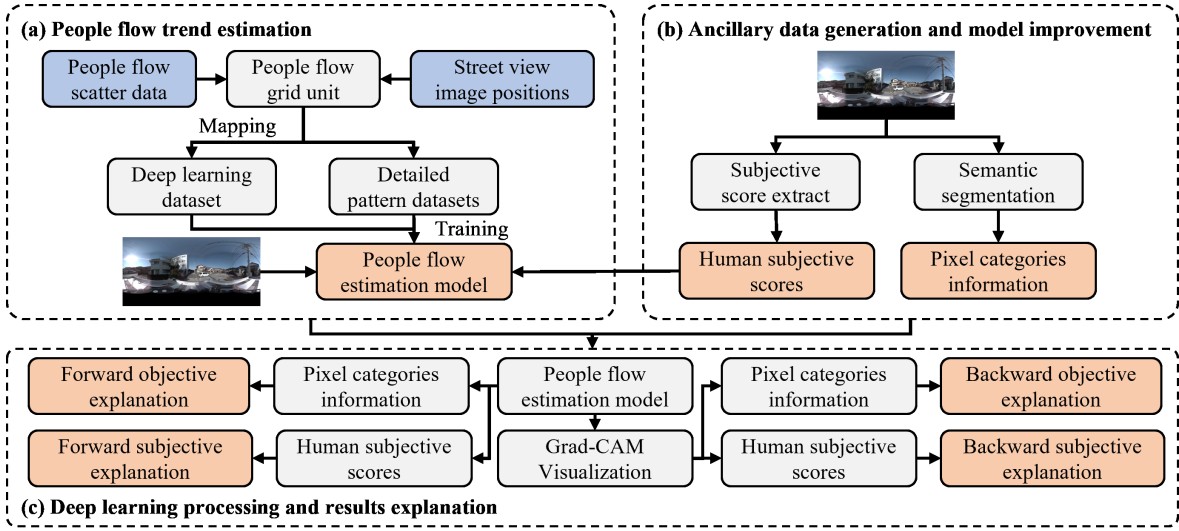

**Figure 1.** Proposed workflow of people flow trend estimation and quantitative deep learning explanation, (**a**) People flow trend estimation; (**b**) Ancillary data generation and model improvement and (**c**) Deep learning processing and results explanation.

### 4.1. People Flow Trend Estimation

Our proposed high semantic level end-to-end people flow trend estimation approach is outlined in Figure 2. First, we converted the people flow data from point to mesh units such that it can cover the entire study area. Second, we created a training dataset by comparing the locations of street view images and the coordinates of mesh units. Finally, we deployed several deep learning backbone algorithms on the street view images and selected the best algorithm to generalize the data to wide area data and data for different time and movement patterns.

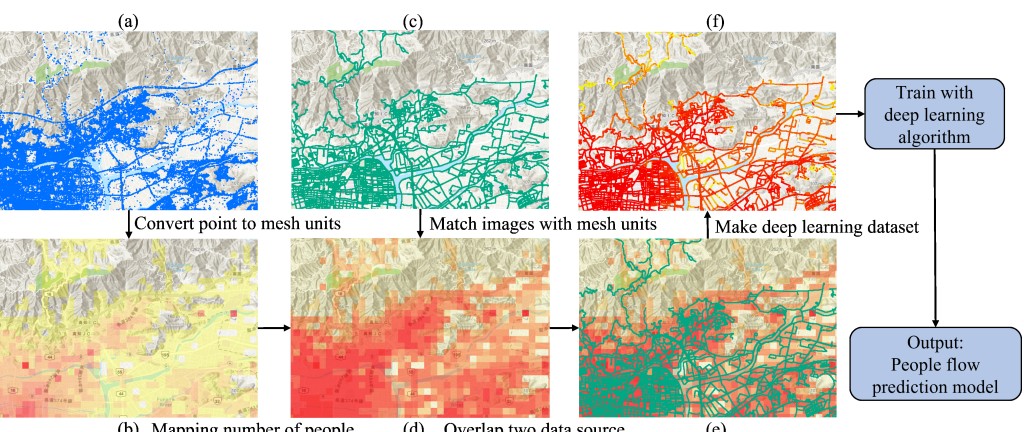

**Figure 2.** Flowchart of people flow trend estimation, (**a**) people flow point raw data; (**b**) people flow mesh data; (**c**) positions of captured street view images; (**d**) mapped people flow mesh data; (**e**) overlapping (**c**,**d**); (**f**) deep learning dataset visualization.

#### 4.1.1. Data Preparation

There were 338,781 points in the people flow trend data in the 761.74 km$^2$ study area. To create the deep learning dataset, the point data had to be first converted into mesh units

to cover the entire area. A grid with a 250 m resolution was generated and the points were counted with a scale parameter in each mesh unit (Figure 2a,b). Because the maximum value of the counted results was 195,992 people and most of the mesh units were sparsely populated, such as less than 100 people on the 250 × 250 m$^2$ grid units, we could not observe any apparent difference in Step (b). Therefore, we implemented the log function on the estimation in Step (b) to (d). To construct the deep learning dataset, the captured street view images were matched with these mesh units in Step (c) to (d). Thereafter, we overlapped the street view position with the mesh units in Step (d) to (e), and discretized the value of the mesh units to 10 levels by rounding the output of the log function. As the value range was $[0, 12]$, we defined Classes 0 to 9 based on whether the number of people in a mesh unit $\in (0, e^2], (e^2, e^3], \ldots, (e^9, e^{10}], (e^{10}, \infty)$, such that the annotations of people flow trend index for each street view image could be obtained in Step (e) to (f).

The category distribution of the total 1,523,882 images is shown in Figure 3a. The amount difference between categories could be up to 10 times, which can be adverse for deep learning training. Therefore, we performed class balancing by random selection and additionally maintained the numerical relationship of categories for model robustness (Figure 2b). Finally, we selected 57,545 images (3.78% of the total number of images), which formed the deep learning dataset. The 10 classes of people flow trend level are shown in Figure 2b, which were further split into training, validation, and test datasets with respective shares of 60, 20, and 20%.

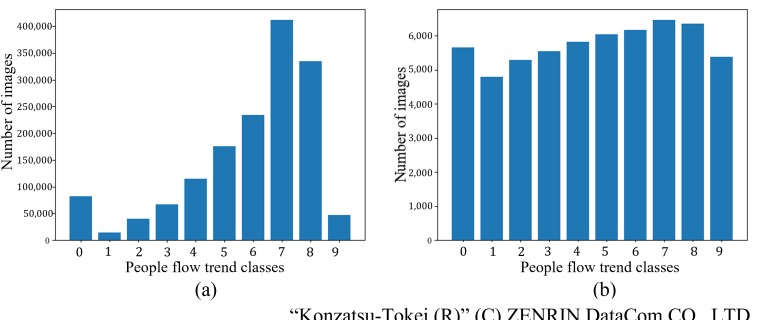

"Konzatsu-Tokei (R)" (C) ZENRIN DataCom CO., LTD.

**Figure 3.** (**a**,**b**) Categories distribution of total image data and deep learning dataset after class balancing.

Previous population mapping studies only focused on the number of people [39,40]; detailed people flow time and movement feature analyses were inadequate. To analyze people flow trend with greater detail and to validate the robustness of the proposed people flow trend, we reprocessed the raw data according to the time and movement feature of the people flow trend generated from the raw data (Figure 4). The raw data were split into stay/move movement patterns, and day/night time patterns (08:00–18:00 and 18:00–08:00, respectively). Thus, we generated four detailed datasets: Day/Stay, Day/Move, Night/Stay, and Night/Move. The dataset visualization and distribution results are shown in Figure 5. The 10 people flow trend levels shown in the histogram were plotted in the map.

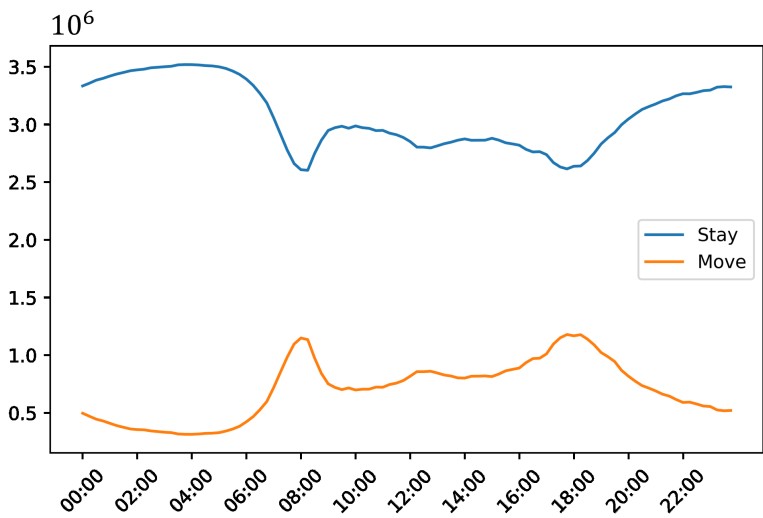

**Figure 4.** Time and movement feature of people flow trend in the study area.

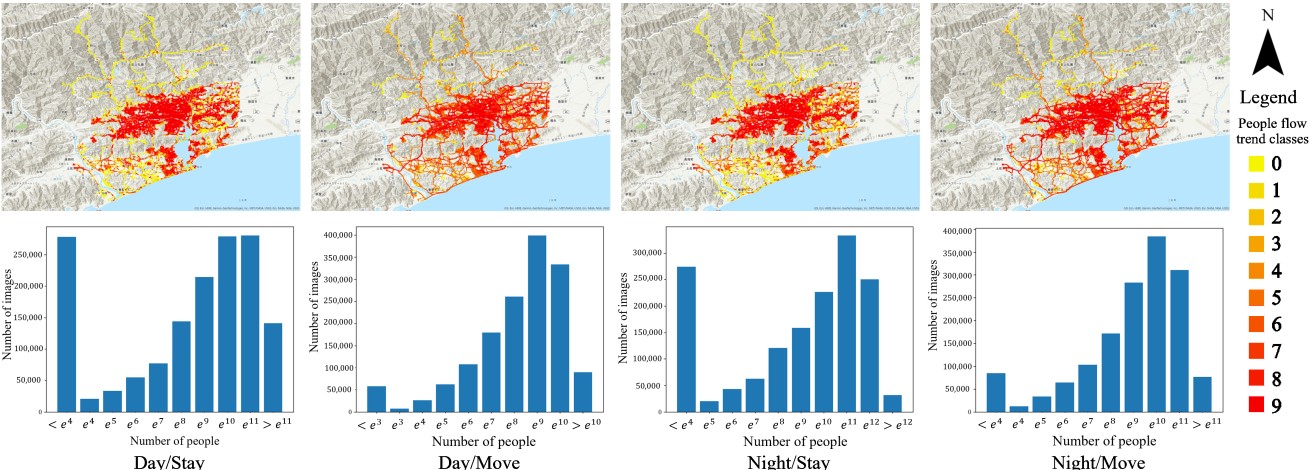

**Figure 5.** Visualization and distribution results of different dataset patterns.

### 4.1.2. People Flow Estimation with Deep Learning Algorithms

The general classification algorithms were trained on over 1 million images labeled across 1000 classes (ILSVRC). They learned the texture and semantic features of this large dataset, which covers most general scenes and objects in our daily life. Therefore, the backbone networks of the classification algorithm could be considered as general feature extractors and transferred to other downstream vision tasks, such as object detection and semantic segmentation. Considering these transfer processes, we proposed an end-to-end approach for estimating people flow trend by transferring the classification backbone models to directly represent the input street scene and the output people flow trend classes. Representative state-of-the-art backbones were selected from CNNs and ViTs separately: the base version of ConvNeXt [41] (ConvNeXt-B), the small and basic versions of the Swin Transformer [42] (Swin-S, Swin-B), along with the set of the old best backbone ResNet as a comparison. The CNNs and ViTs implemented in this study are summarized in Table 1, where the memory and computational requirements were calculated when the input image size was 224 × 224.

Swin Transformer first splits an input image $X \epsilon \mathbb{R}^{H \times W \times C}$ into non-overlapping patches using a patch-splitting module, which is similar to the original ViT. Each patch is treated as a "token" and its feature is set as a concatenation of the raw pixel RGB values. Then, several transformer blocks with modified self-attention computation form the different stages and are applied to these patch tokens. Similar to the ResNet bottleneck and skip

connection architecture, tokens are downsized as the stages deepen. Unlike the original ViT, the Swin transformer alternately applies window-based MSA (W-MSA) and shifted window-based self-attention (SW-MSA) in the Swin transformer block as follows:

$$
\begin{aligned}
\hat{z}^l &= \text{W-MSA}\left(LN\left(z^{l-1}\right)\right) + z^{l-1}, \\
z^l &= MLP\left(LN\left(\hat{z}^l\right)\right) + \hat{z}^l, \\
\hat{z}^{l+1} &= \text{SW-MSA}\left(LN\left(z^l\right)\right) + z^l, \\
z^{l+1} &= MLP\left(LN\left(\hat{z}^{l+1}\right)\right) + \hat{z}^{l+1}
\end{aligned}
\tag{1}
$$

where W-MSA applies windows that contain $M \times M$ patches and SW-MSA shift window and permutation feature maps to ensure efficient information exchange in self-attention sections; the self-attention in the Swin transformer is computed as

$$
Attention(Q, K, V) = Softmax\left(\frac{QK^T}{\sqrt{d}} + B\right)V
\tag{2}
$$

where $Q, K, V \in \mathbb{R}^{M^2 \times d}$ represent the query, key and value matrices, $d$ represents the query/key dimension, and $M^2$ represents the number of patches in a window. The head part of the Swin transformer, i.e., the FC layers and loss functions, are similar to the DCNNs and original ViT. ConvNeXt is a variant of ResNet. Considering the same memory and compute consumption, ConvNeXt small and base version are the improvement of ResNet 50 and 200. ConvNeXt was propose in 2022 while ResNet was 2015, Liu et al., considered most of the improvement architectures on ResNet in these 7 years and assimilated the useful ones to construct ConvNeXt. (1) improved training techniques were deployed on ResNet. (2) adjust the number of blocks in each stage to align the floating point operations (FLOPs) with Swin Transformer. (3) Modified the first convolution layer to the patch processing of Swin Transformer. (4) Modified the general convolution layers to depthwise convolution. (5) Tuning the channel amount and patch sequence. (6) Change the general $3 \times 3$ convolution kernel to 5, 7, 9, 11, and get the best performance when the kernel size is $7 \times 7$. (7) Change the activation function from ReLU to GELU. (8) Reduce the use of batch normalize (BN) layer and the activation function. (9) Substituting BN with LN. (10) Change the down sampling process from $3 \times 3$ convolution with stride 2 to $2 \times 2$ convolution with stride 2. By these modifications, ConvNeXt was constructed from ResNet.

**Table 1.** Summary of memory and computation requirements of deep learning people flow estimation networks.

| Networks | Params (M) | FLOPs (G) |
|---|---|---|
| ResNet-101 | 44.55 | 7.85 |
| Swin transformer small | 49.61 | 8.52 |
| Swin transformer base | 87.77 | 15.14 |
| ConvNeXt base | 88.59 | 15.36 |

*4.2. Ancillary Data Generation and Model Improvement*

We implemented and an improved a human subjective score extraction approach with respect to street view images, and deployed this approachit on the images to provide ancillary data for improving the people flow estimation model improvement and for the third part of the explanationanalysis. Furthermore, we deployed semantic segmentation algorithm on the street view images to obtain the pixel-level category information additionally as further ancillary data to increase for increasing the detail of analysis in the third component.

4.2.1. Subjective Score Extraction Dataset

For the detailed analysis of the deep learning explanation component and the improvement of the subjective score extraction performance, our dataset was marginally different from the study by Dubey et al. We selected 1500 representative panoramic street view images and constructed 14,950 pairs. For the image pairs setting, we selected images which were captured in 10 types of different land use area defined by Japanese government. (Category I exclusively low-rise, Category II exclusively low-rise, Category I mid/high-rise oriented residential zone, Category II mid/high-rise oriented residential zone, Category I residential zone, Category II residential zone, Quasi-residential zone, Neighborhood commercial zone, Commercial zone, Quasi-industrial zone). Then we created crossing comparison among these 10 land use types, we have 20 comparisons per image per question on average, better than 3.35 in [38]. In each pair, we surveyed all the 22 perceptual attributes, shown as Figure 6, rather than surveying randomly selected items from the six perceptual attributes. In these 22 perceptual attributes, we set some similar (Comfortable/Neat, Attractive/Like, etc.) and opposite questions (Depressing/Like, Lively/Boring) to verify the internal rationality of our subjective score extraction model, the detailed explanation is illustrated in Section 5.2.1. Furthermore, for each perceptual attribute, we conducted a survey using 40 volunteers. In total, 38,525 volunteers were surveyed in this study. Therefore, the total volume of our dataset was $14{,}950 \times 22 \times 40 = 13{,}156{,}000$. We split the 14,950 pairs into training, validation, and test datasets with respective shares of 60, 20, and 20%.

**Questionnaire**

Which image looks more

0.Open, 1.Friendly, 2.Lively, 3.Comfortable, 4.Greenery, 5.Calm, 6.Bright, 7.Old-fashioned, 8.Safe, 9.Neat, 10.Lived-in feel, 11.Cosy, 12.Clean, 13.Beautiful, 14.Wealthy, 15.Boring, 16.Depressing, 17.Like, 18.Interesting, 19.Desireable for living, 20. Desirable for going through, 21.Attractive

Image 1

Image 2

**Figure 6.** Survey to construct subjective score extraction dataset.

4.2.2. Human Subjective Score Extraction and People Flow Estimation Model Improvement

As we possessed 14,950 image pairs and emotion comparations, covering all the 22 perceptual attributes on each pair, we could convert the human subjective score extraction task from the original single logit value output to a multi-labeled classification. The architecture is shown in Figure 7.

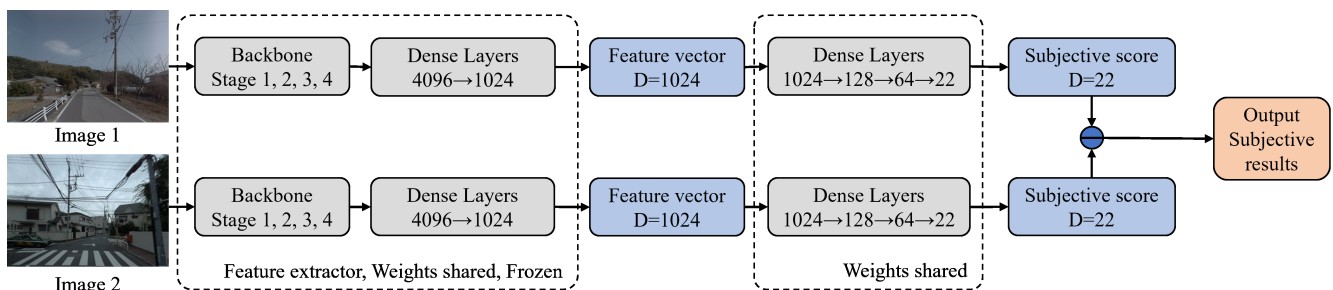

**Figure 7.** Schematic of subjective score extraction model.

The multi-labeled classification model is Siamese-like deep learning architecture, this type of architecture has been used in many tasks that focus on change or comparation [43,44]. Thus, it is appropriate for comparing the subjective feeling between two images. First, we cropped the center part of the panoramic images, and rectified the lens distortion. We then separately input two images to a weight-frozen deep learning backbone that was pretrained by ImageNet dataset, to obtain two 1024-dimensional vectors, extracted

by this backbone. Second, we input these two vectors to weight shared dense layers to obtain two 22-D vectors, which indicate the subjective scores. Finally, we subtracted the 22-D vector of image 2 from the vector of image 1 to obtain a 22-D subjective result, that is, the answer to whether Image 1 looks more open/friendly/lively . . . than Image 2 on these 22 perceptual attributes. The loss value was calculated as the binary cross entropy (BCE) between the 22-D results and the questionnaire results of these 22 perceptual attributes.

After the training converged, as two branches of this Siamese network were weights shared, any one of the branches could be considered as the quantitative subjective score extractor. Additionally, we improved the proposed people flow estimation model, based on the subjective scores extracted from one branch of the Siamese network (Figure 7). The architecture of the people flow trend estimation improvement is shown in Figure 8.

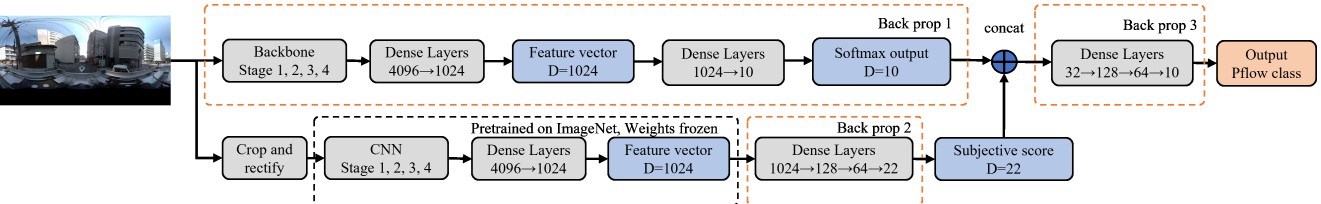

**Figure 8.** Architecture of the people flow trend estimation improvement.

For this improvement, we considered the 22-D subjective score of each image as ancillary data, which can positively influence the people flow trend estimation improvement task. First, we input an image to the original people flow estimation network to obtain the output of the Softmax function before classification, which can be considered as the first back propagation phase in the training. Second, we cropped the center part of the panoramic images and rectified them to input them to the subjective score extractor. Thus, we obtained the 22-D subjective scores. The dense layer of this extractor was the second phase back propagation in the training. Finally, we concatenated the 22-D subjective scores with the 10-D Softmax output, and these 32-D vectors were input to the final phase back propagation dense layer. The obtained output was the 10-D improved people flow trend estimation result.

### 4.2.3. Pixel Level Categories Information Extracted through Semantic Segmentation

To obtain detailed information for the model and estimation result explanation, we implemented the semantic segmentation branch of the unified perceptual parsing network (UperNet) [45] on the street view images to provide pixel level categories information for further analysis.

In our implementation, as the segmentation model pretrained on the open street view datasets functioned effectively on our dataset without fine tuning, and because we require more detailed pixel level categories information for further analysis, we selected a state-of-the-art model that was pretrained on ADE20K [46] open dataset, which contains 150 segmentation classes, for which the backbone was Swin-B.

### 4.3. Explanation of Deep Learning Processing and Results

#### 4.3.1. Forward Objective and Subjective Explanation

Ten discrete classes were obtained as results from the proposed people flow trend estimation. For the forward explanation, we needed to understand the pixel distribution feature and human subjective impression feature of each class. For the objective analysis, we separately calculated the pixel proportion of each segmentation class for all 10 people flow classes, and for all different patterns of Day/Stay, Day/Move, Night/Stay, Night/Move. Subsequently, we selected the segmentation classes of the top 20 pixel proportions to normalize and visualize them. For the subjective analysis, we separately calculated the mean values of the 22 subjective scores for all the 10 people flow classes, and in the four

time and movement patterns. Finally, we normalized and visualized the 22 mean values of the 10 people flow classes.

4.3.2. Implementation of Grad-CAM and Proposal of Gradient Impact Method

Grad-CAM is a method to visualize the attention of deep learning networks. Its output includes the attention scores of each pixel, that is, it can provide information with respect to the image exerting the greatest impact on the results. Thus, in our study, we implemented Grad-CAM and combined the output with our pixel category information ancillary data to quantitatively analyze the pixel categories exerting the greatest impact on the people flow trend estimation. The architecture of Grad-CAM is schematically depicted in Figure 9.

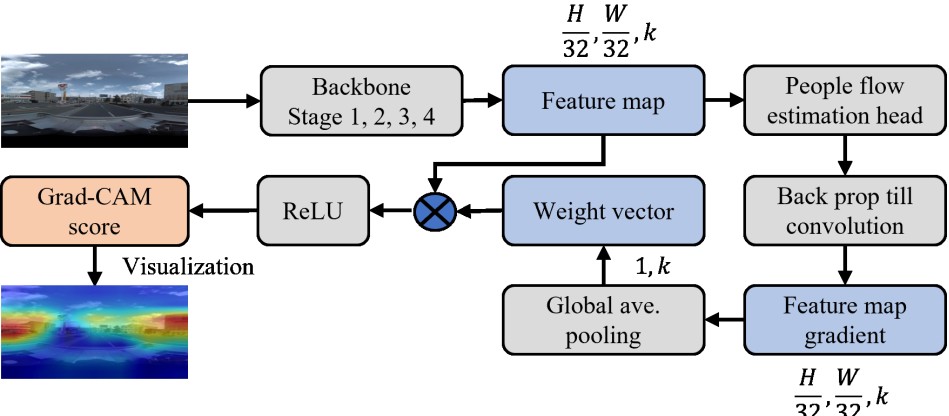

**Figure 9.** Architecture of Grad-CAM.

Provided an image and a people flow estimation class as input, we forward propagated the image through the backbone component of the model and subsequently through task-specific computations to obtain a raw score for the category. The gradients were set to 0 for 9 classes, whereas the desired class was set to 1. This signal was then back propagated to the output of the backbone to calculate the feature map gradient as follows:

$$\alpha_k^c = \overbrace{\frac{1}{Z}\sum_i\sum_j}^{GAP} \underbrace{\frac{\partial y^c}{\partial A_{ij}^k}}_{gradients} \tag{3}$$

where $y^c$ is the raw score for the specific one class output from the people flow estimation head shown in Figure 9; $c$ is the specific class; $A_{ij}^k$ is the feature map output from the backbone; $i$ and $j$ indicate the position of the feature map; $k$ is the feature map channel amount; the gradient of each position was processed through a global average pooling (GAP) layer; and $\alpha_k^c$ is the weight vector and was deployed on the feature map as follows:

$$L_{Grad-CAM}^c = ReLU\left(\underbrace{\sum_k \alpha_k^c A^k}_{linear\ combination}\right) \tag{4}$$

where the $L_{Grad-CAM}^c$ is the localization Grad-CAM score, which is obtained as the output of Grad-CAM; the shape is obtained as $\left(\frac{W}{32}, \frac{H}{32}, 1\right)$; and the total output is $L_{Grad-CAM}$, with shape $\left(\frac{W}{32}, \frac{H}{32}, n\right)$, where $n$ is the number of images used. Grad-CAM calculated the gradient of the backbone output feature map and deployed the gradient on the feature map to calculate the score, and finally aligned it with the original image. Thus, the high

Grad-CAM score position refers to the pixels in that corresponding position that exerts the greatest impact on the people flow estimation results. Following this principle, we proposed the gradient impact method to obtain the subjective score exerting the greatest impact on the people flow estimation result, which is provided by Equation (5) as follows:

$$L^c_{Grad-IMP} = ReLU\left(\frac{\partial y_c}{\partial x}\right), \; y = Net(x) \tag{5}$$

where $x$ is the input of back propagation 3 shown in Figure 8, that is, for the 32-D vector; Net represents the back propagation 3; $y$ is the logit value before classification; $c$ is the class of people flow estimation; $L^c_{Grad-IMP}$ is the gradient impact score, which represents the importance of the 32-D vector separately, similar to $L^c_{Grad-CAM}$, with shaped $(1, n)$.

### 4.3.3. Backward Objective and Subjective Explanation

Similar to the process described in Section 4.3.1 for the backward explanation, the impact of pixel distribution feature and human subjective score to each people flow class needs to be analyzed. For the objective analysis, we separately calculated the Grad-CAM score mean value that was resized and aligned to each segmentation class in these ten people flow classes as shown in Equation (6), together with the patterns of Day/Stay, Day/Move, Night/Stay, Night/Move.

$$L^s_{Grad-CAM} = \frac{1}{p}\sum_{i,j}^{p} f_a(L_{Grad-CAM}) \tag{6}$$

where $f_a$ is the aligning function, which aligned the $L_{Grad-CAM}$ to the segmentation output. In addition, the alignment was deployed after cropping the bottom part of the image, primarily containing only the vehicle itself, based on which the subsequent analyses were performed. $(i, j)$ represents the pixel position, which belongs to the s class; $p$ is the pixel amount, which belongs to the $s$ segmentation pixel class; $L^s_{Grad-CAM}$ is the score of $s$; the shape was $(1, n)$. Thereafter, we aggregated each $L^s_{Grad-CAM}$; thus, the shape was $(150, n)$, where 150 is the segmentation pixel class of the ADE20K dataset. Finally, we calculated the mean value according to the 10 people flow class, and the final output of backward objective analysis was termed $L^{obj}_{Grad-CAM}$, with shape $(150, 10)$.

As the parameters we require for the subjective analysis were just the importance values of the 32-D vector, it was considerably simpler than objective analysis. We aggregated the $L^c_{Grad-IMP}$ of the 32-D vector, with shape $(32, n)$, and additionally calculated the mean value according the 10 people flow class. The final output of the backward subjective analysis was named $L^{sbj}_{Grad-IMP}$, shape $(32, 10)$.

## 5. Experiments and Results

Our experiment was deployed on a cloud platform called mdx [47], which provides users with flexibility, strong security, and the ability to couple with supercomputers and edge devices, through high performance networks. The virtualization service of the mdx platform, jointly operated by nine national universities and two national research institutes in Japan, was launched in 2021. The proposed deep learning architectures were primarily processed on four Nvidia A100 40G.

### 5.1. People Flow Trend Estimation

#### 5.1.1. Image Data Pre-Processing

The street view images and corresponding distribution (Figure 3b), with the discretized classes are shown in Figure 10. The dataset was split into training, validation, and test datasets with the respective shares of 60, 20, and 20%. We then performed our test on a wide area source dataset. Thereafter, the people flow trend estimation task could be designed as a general classification workflow.

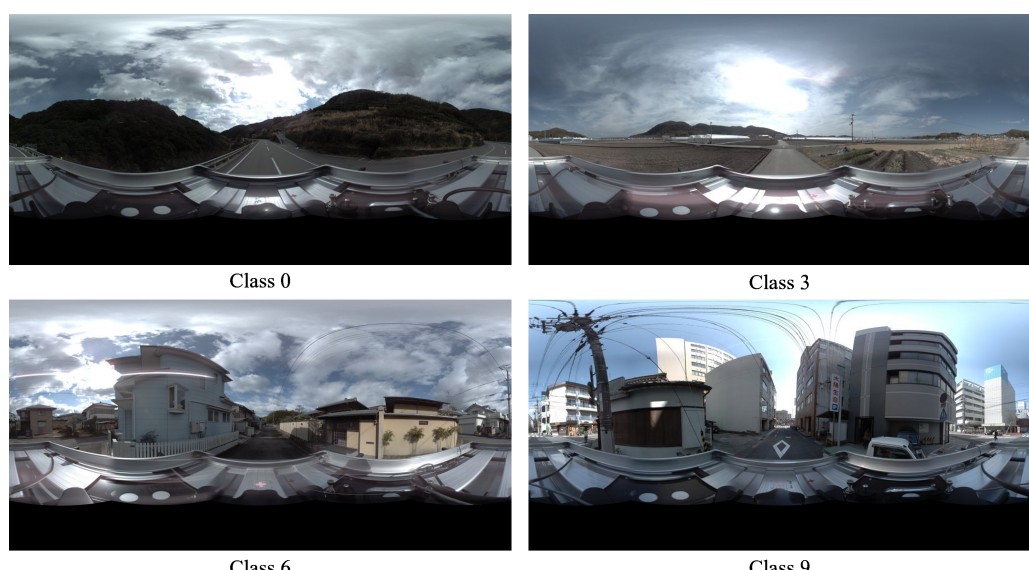

**Figure 10.** Examples of the street view images corresponding to the discretized classes.

5.1.2. Results of People Flow Trend Estimation

We implemented ResNet101, Swin-S, Swin-B, and ConvNeXt-B to compare the old CNN benchmarks of the ViT and CNN state-of-the-art architectures, and the models were implemented for only a portion (3.78%) of the total dataset, and performed a precision estimate on the test set. Thereafter, we selected Swin-B and ConvNeXt-B to perform the comparison on the total wide data of approximately 1.5 million images. We selected precision, recall, mF1, and accuracy as our evaluation index, and the results are summarized in Table 2. In addition, we visualized the wide-area inference result and compared it with the ground truth (Figure 11).

**Table 2.** Classification performance of different experiments.

| Approach | Recall | Precision | mF1 | Accuracy |
|---|---|---|---|---|
| ResNet-101 | 0.4974 | 0.4888 | 0.4898 | 0.4894 |
| Swin-S | 0.7272 | 0.7242 | 0.7255 | 0.7174 |
| Swin-B | 0.7676 | 0.7669 | 0.7672 | 0.7584 |
| ConvNext-B | 0.7904 | 0.7883 | 0.7892 | 0.7812 |
| Swin-B, total | 0.7802 | 0.7059 | 0.7367 | 0.7106 |
| ConvNext-B, total | 0.7981 | 0.7211 | 0.7528 | 0.7271 |

The accuracy metrics mentioned in Table 2 were calulated as Equation (7)

$$
\begin{aligned}
Precision &= \frac{TP}{TP + FP}, \\
Recall &= \frac{TP}{TP + FN}, \\
F1 &= \frac{2 \times precision \times Recall}{Precision + Recall}, \\
Accuracy &= \frac{TP + TN}{TP + TN + FP + FN}
\end{aligned}
\tag{7}
$$

Figure 11a–c show the wide area prediction results, the wide area ground truth with legend, and the comparison of prediction with ground truth (prediction-ground truth). Although many images were incorrect, they mostly occurred in proximity to the diagonals

of the confusion matrices. Therefore, it can be observed that we constructed an effective people flow trend prediction model.

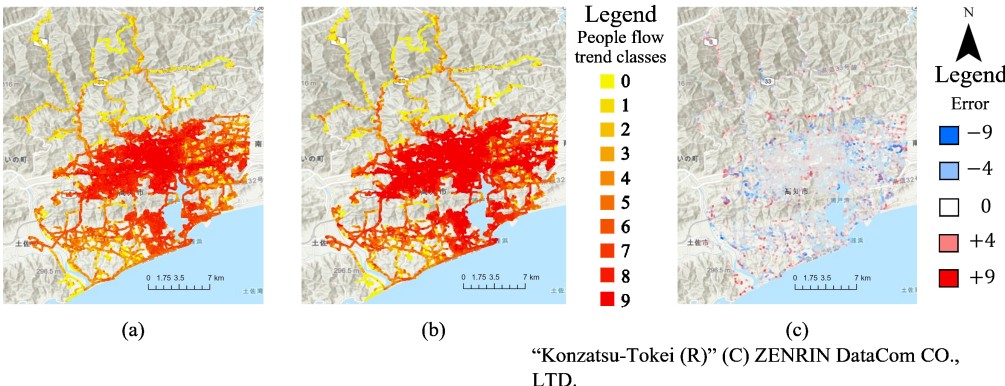

**Figure 11.** Visualization of people flow trend in wide area estimation result, (**a**) wide area prediction results; (**b**) the wide area ground truth with legend; (**c**) the comparison of prediction with ground truth.

The pre-processing and the training strategies adopted for the people flow trend estimation for the detailed Day/Stay, Day/Move, Night/Stay, Night/Move patterns were the same as those adopted for the wide area testing, and the results are presented in Table 3. We selected the best model among the aforementioned experiments by deploying the ConvNeXt-B algorithm on these four datasets.

**Table 3.** Classification performance of different patterns.

| Object | Recall | Precision | mF1 | Accuracy |
|---|---|---|---|---|
| Day/Stay | 0.6544 | 0.6566 | 0.6552 | 0.6444 |
| Day/Move | 0.7147 | 0.7154 | 0.7146 | 0.7027 |
| Night/Stay | 0.6983 | 0.6988 | 0.6983 | 0.6866 |
| Night/Move | 0.7093 | 0.7113 | 0.7100 | 0.6996 |

Table 2 shows that ConvNeXt exhibited the highest performance of 78.12 and 72.71% on the deep learning dataset and wide area data, respectively. Table 3 shows the accuracies of Day/Stay, Day/Move, Night/Stay, Night/Move patterns to be 64.44, 70.24, 68.66, and 69.96% respectively. As we only selected 3.78% images as the training, validation, and testing datasets, if we deployed our estimation model on the wide area data, the accuracy would be inferior.

### 5.1.3. Comparison of Discrete Classification and Continuous Regression

Intuitively, as the people flow trend value is a continuous value, the people flow estimation can be considered as a continuous regression task. However, in some cases, continuous ground truth is too precise, which renders the mapping of the image to the precise continuous ground truth difficult. Furthermore, performing the gradient descent with just the constraint of L1 loss is difficult, and many regression tasks require more than one refinement processes, such as two stage object detection algorithms [48–50].

To overturn the intuitive assumption and prove the superiority of the proposed selection of classification, we used the data as the ground truth prior to discretizing for implementing the regression directly as the comparation. The data pre-processing and training strategies were approximately the same as those discussed in Section 5.2.1. The change of the classification head to a regression head in the network was the only modification, which changed the loss from cross entropy loss to L1 loss. We selected Swin-B

as the comparative experiment backbone network. The comparation results are shown in Figure 12.

Figure 12a shows the regression results on the test set, and the L1 loss were calculated as

$$L_1 = \frac{1}{n} \sum_{i=1}^{n} |f(x_i) - y_i| \tag{8}$$

where $x_i$ is the $i$ th image, $f$ is the deep learning algorithm, $y_i$ is the $i$ th ground truth, and $n$ is the volume of the test set. This regression L1 test loss in Figure 12a is 0.7798. Figure 12b shows the confusion matrix of converting the regression back to discrete classification; the accuracy was only 43.25%, which is significantly worse than the original classification accuracy (75.84%). Thus, it is better to deploy the strategy based on classification task rather than direct regression in this study.

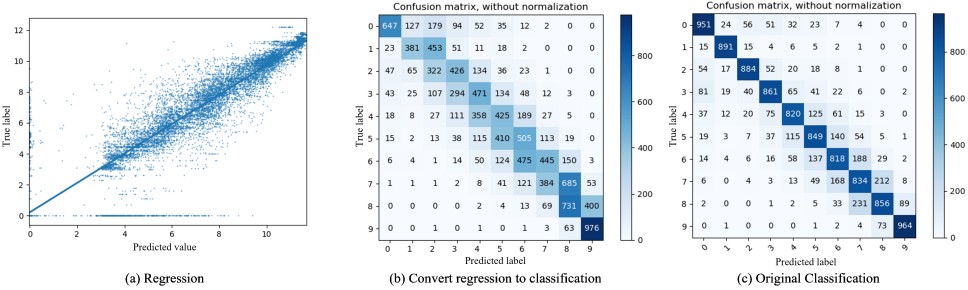

(a) Regression     (b) Convert regression to classification     (c) Original Classification

**Figure 12.** Comparison of discrete classification and continuous regression.

### 5.2. Ancillary Data Generation and Model Improvement

5.2.1. Siamese Network Training

To extract the human subjective scores of the 22 perceptual attributes discussed in Section 4.2.1, we trained the Siamese (Figure 7), to analyze the comparation relationship of the image pairs. The shape output of the Siamese network was $(n, 22)$, where n is the number of images. These output indicate whether Image 1 appears more open/genial/lively ... than Image 2 on these 22 perceptual attributes, the 22 perceptual attributes were shown in Figure 6.

For the training, we compared the original network architecture [38] of the VGG-16 backbone and the single labeled classification with our proposed improvement of the ConvNeXt-B backbone and multi-labeled classification. We set the learning rate as 0.0001 and trained 100 epochs for each experiment. The comparation results are shown in Figure 13.

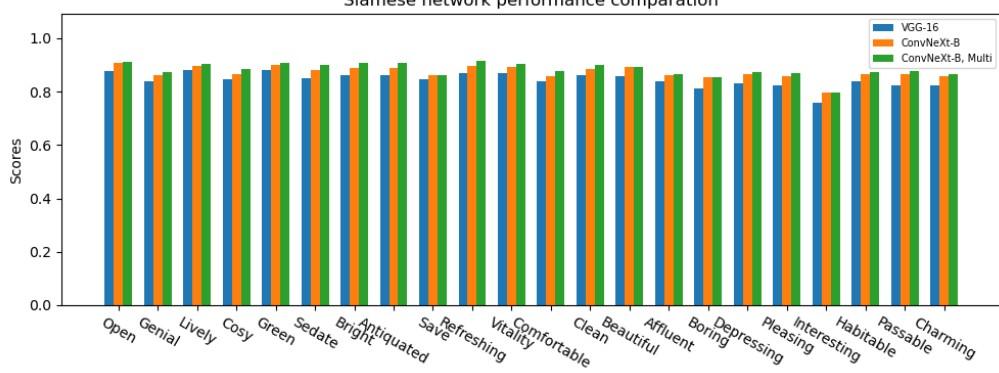

**Figure 13.** Siamese network performance comparison.

As observed, the proposed improvement achieved the highest performance; the overall accuracy of the VGG-16, and ConvNeXt-B and multi-labeled classification were 84.46, 87.24,

and 88.23%. Thus, the subjective score extracted from the branch of ConvNeXt-B and multi-labeled classification can be considered as the most reliable; we deployed this model on the next experiments.

For validating the internal rationality of this subjective score extraction algorithm, we calculated the correlation coefficients matrix of the subjective perception scores (Figure 14). Theoretically, correlation coefficients between similar perceptions will be higher, otherwise they will be lower. The matrix shows correlation coefficients between similar perception as Comfortable/Neat (0.84), Attractive/Like (0.97) are high; Depressing/Like (−0.96), Lively/Boring (−0.84) are low. The score extracted by our model are statistically reasonable.

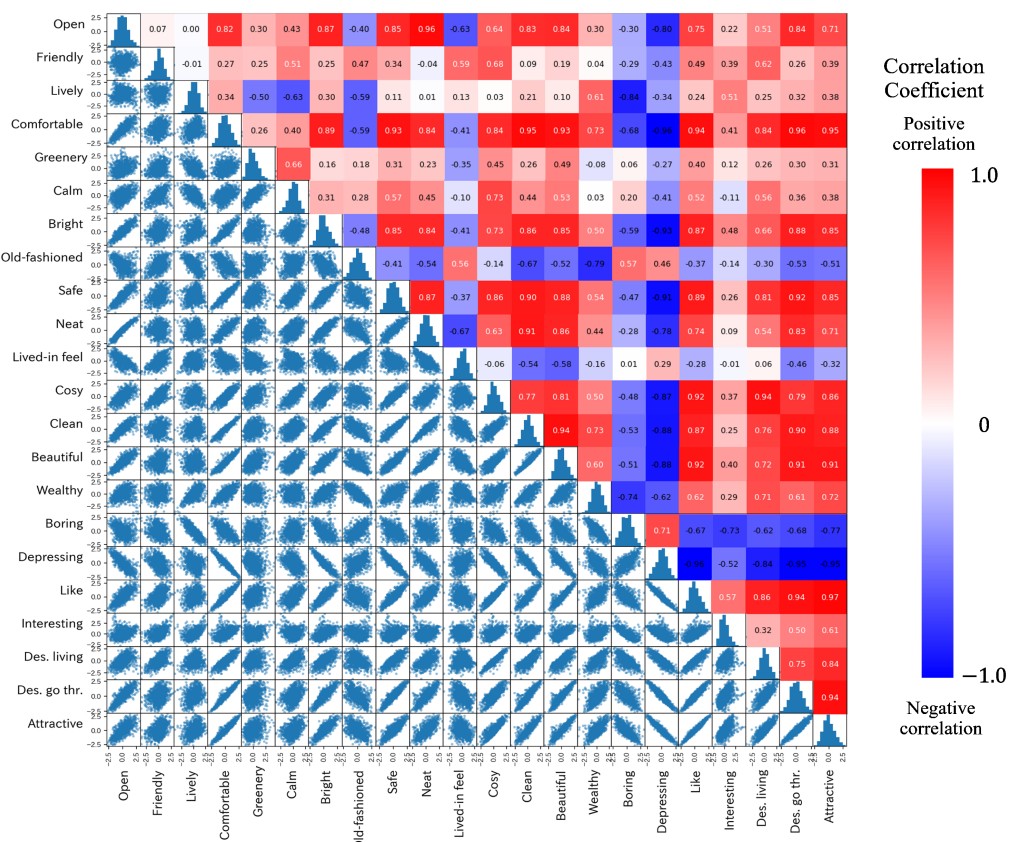

**Figure 14.** Coefficients matrix of the subjective perception scores.

### 5.2.2. Human Subjective Score Extraction and People Flow Estimation Model Improvement

We split out one branch discussed in Section 5.1.1, and input the center cropped and rectified the street view images dataset discussed in Section 4.2.1; thereafter, we extracted a $(n, 22)$ shaped score, where $n$ is the number of images. Moreover, we know that the logit value output of the people flow estimation model was shaped as $(n, 10)$. Thus, the $(n, 22)$ shaped score can act as the ancillary data to improve the people flow estimation performance. We concatenated the logit value output and the subjective score which yielded a $(n, 32)$ shaped input. Subsequently, $(n, 32)$ was input to the network described in Figure 8. The improved performance of the people flow trend estimation is shown in Table 4, which can be compared with Tables 2 and 3.

We can observe that the subjective score ancillary data improved the people flow estimation performance compared with that of the method that only used the street view images as training data.

**Table 4.** Improved people flow trend estimation performance.

| Original + Subjective | Recall | Precision | mF1 | Accuracy |
|---|---|---|---|---|
| Overall | 0.7921 | 0.7909 | 0.7914 | 0.7832 |
| Day/Stay | 0.6559 | 0.6572 | 0.6561 | 0.6462 |
| Day/Move | 0.7155 | 0.7194 | 0.7169 | 0.7040 |
| Night/Stay | 0.6995 | 0.7015 | 0.6999 | 0.6880 |
| Night/Move | 0.7109 | 0.7132 | 0.7117 | 0.7012 |

5.2.3. Pixel Level Categories Information Extracted by Semantic Segmentation

As discussed in Section 4.2.3, we deployed the semantic segmentation algorithm on the street view images. We inferenced the images using UperNet and Swin-B backbone pretrained on the ADE20K dataset. The results provided pixel-level category information for the subsequent experiments.

*5.3. Explanation of Deep Learning Processing and Results*

5.3.1. Forward Objective and Subjective Explanation

The output shape of the segmentation discussed in Section 5.2.3 is $(n, w, h, 1)$, where n is the number of images; $w$ and $h$ are the width and height of the image, respectively; the last channel indicates the pixel segmentation classes, in the ranged $[0, 149]$. To analyze the relationship between the street view scenes and people flow trend, we calculated the pixel proportion of each segmentation class, which yielded the shape $(n, 150)$. Thereafter, we calculated the proportion mean values according to the people flow trend class, which yielded the shape $(10, 150)$. We then selected the top 20 segmentation class proportions to visualize the relationship between scene characteristics and people flow trend. The visualization results of the Day/Stay, Day/Move, Night/Stay, and Night/Move patterns are shown in Figure 15. They were the normalized proportions. As the people flow trend level increased, the proportion of the natural pixels such as mountain and tree decreased, whereas the artificial pixels such as buildings and roads increased. In addition, some segmentation classes such as plants and ships did not follow these rules, because people may also aggregate in areas such as city parks with many plants and the ship can be only observed from the coastal area with specific people distributions.

We analyzed the objective factors such as segmentation result, in addition to the relationship between subjective scores and people flow trend. The output shape of the subjective extraction discussed in Section 5.2.2 was $(n, 22)$. We also calculated the subjective score mean values according to the people flow trend class, which yielded the shape $(10, 22)$. The visualization results of the four patterns are shown in Figure 16. The visualization results were the normalized scores, where the subjective score class was associated to the class described in Figures 6 and 19. As the people flow trend level increased, the positive subjective scores such as Sub 1 and Sub 2, that is, friendly and lively, increased and the negative scores such as 15 and 16, that is, boring and depressing, decreased. The subjective score Classes 0 and 4 did not follow these rules, because areas such as the downtown area appears not sufficiently open and green. Moreover, the subjective score distribution was more concentrated in the Stay patterns than in the Move pattern.

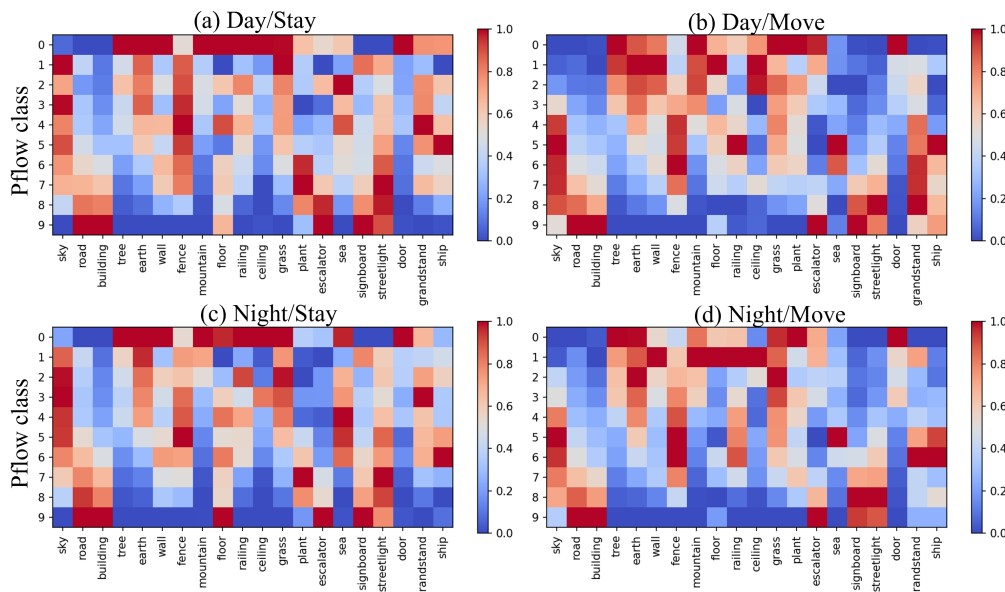

**Figure 15.** Relationship between scene characteristics and people flow trend.

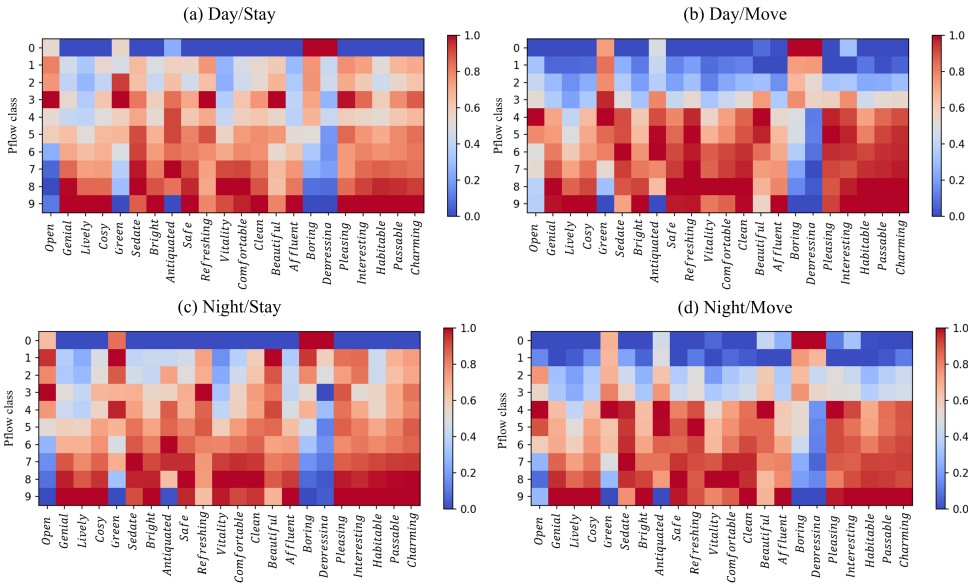

**Figure 16.** Relationship between subjective scores and people flow trend.

Moreover, Figure 5 shows the similar people flow distribution characteristics of the four patterns: It indicates that people tended to concentrate in the city area rather than countryside. The objective pixel distribution and subjective score characteristics in different people flow levels were similar in these four patterns: people tended to aggregate in the scene with larger artificial pixel proportions and higher positive subjective score. In addition, Figure 5 shows that people flow in the Stay pattern was more concentrated than in the Move pattern, whereas the distributions of the Day/Night patterns (concentrated) were similar, which rendered the relationship between the objective/subjective factors and the people flow trend appear more concentrated in the Stay pattern than in the Move pattern and similar in the Day/Night patterns.

### 5.3.2. Grad-CAM Implementation

Based on the principle discussed in Section 4.3.2, the score of Grad-CAM is the gradient with the last feature map of the backbone, which indicates the attention extent of the deep

learning algorithms. However, the raw output was shaped as $\left(\frac{W}{32}, \frac{H}{32}, 1\right)$ for each image. To understand the people flow estimation model proposed in this study with greater clarity, we visualized the Grad-CAM score. We resized the $\left(\frac{W}{32}, \frac{H}{32}, 1\right)$ score back to $(w, h, 1)$, and converted it to the heatmap to overlap with the original image. The visualization examples are shown as Figure 17, where the backbone network based on the Grad-CAM shown in Figure 9, we selected is the ConvNeXt-B.

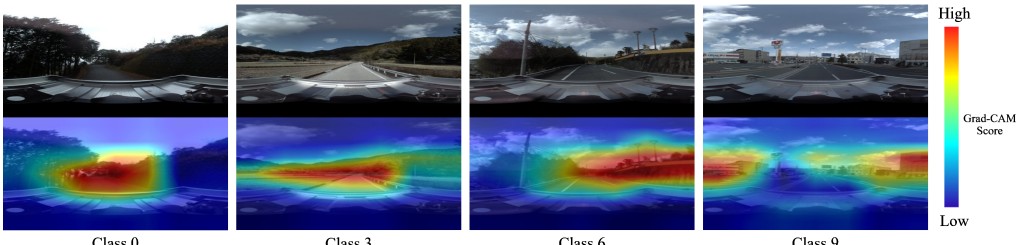

**Figure 17.** Examples of Grad-CAM visualization.

The visualization results show that in low people flow trend classes such as Classes 0–3, the attention of estimation algorithm tended to concentrate on the pixels belonging to the natural categories such as mountain, tree, and earth. In higher people flow trend classes such as Classes 6–9, the attention tended to concentrate on the pixels belonging to artificial categories such as building, wall, and road. Previous studies have also implemented Grad-CAM on the models on the street view images to visualize the results [36,51,52]. However, it would be superficial to stop at the visualization. Therefore, we further analyzed the raw Grad-CAM scores quantitatively.

### 5.3.3. Backward Objective and Subjective Explanations

We quantitatively analyzed the Grad-CAM score with the pixel categories information as discussed in Section 4.3.3. The result was $L_{Grad-CAM}^{obj}$, shaped as $(150, 10)$.

The attention values in the natural and artificial categories were negatively and positively correlated, respectively with the people flow trend. In addition, we calculated the average value of Grad-CAM scores and gradient impact scores, according to the pixel categories information to obtain the pixel class and the perceptual attributes that exert the greatest impact on the estimation model. Thus, the $(150, 10)$ shaped $L_{Grad-CAM}^{obj}$ discussed in Section 4.3.3 was converted to $(150, 1)$; and the $(32, 10)$ shaped $L_{Grad-IMP}^{sbj}$ was converted to $(32, 1)$; the backward objective and subjective explanation analysis results for the four patterns were shown as Figure 18. As some of the 150 segmentation classes only shared marginal proportions of the entire images, we selected only the classes in the top 20 Grad-CAM scores from the 150 classes.

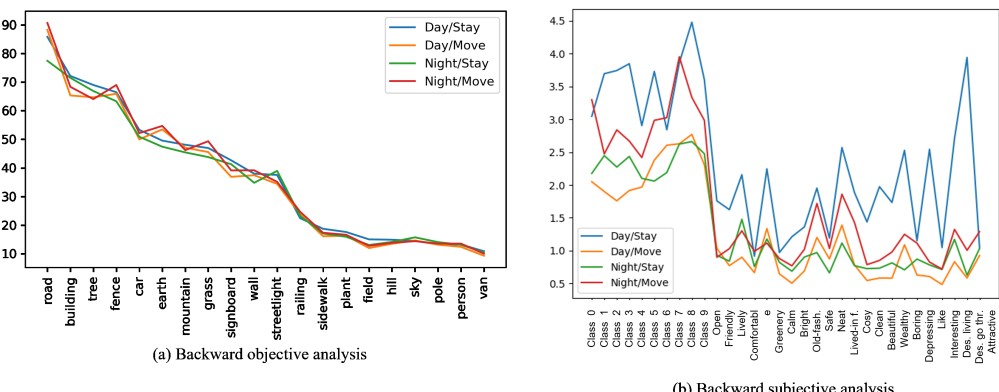

**Figure 18.** Backward objective and subjective explanation results.

The objective analysis is shown in Figure 18a. The categories such as road, building, tree, and fence exerted the highest impact on the people flow trend estimation model. The subjective analysis is shown in Figure 18b. The Classed 0–9, that is, the logit value output from the people flow estimation model exerted the greatest impact. Excepting these 10 logit values, the classes such as Sub 19 (Desirable for living), Sub 10 (Lived-in feel), and Sub 4 (Greenery) show the greatest impact on the people flow estimation.

### 5.3.4. Rationality Analysis of Backward Objective and Subjective Explanations

To prove the reliability of the backward objective and subjective explanations, we implemented a more statistically explainable sparse model (SpM) regression approach on the people flow trend estimation task: the least absolute shrinkage and selection operator (LASSO) [53]. This SpM can simultaneously optimize the objective function and select the parameters that show the greatest impact on the model. Previous studies have implemented this regression approach, considering the weight as the importance of the feature to the regression model [54–57]. The input of our the SpM is the pixel proportion of each image that were output by the semantic segmentation discussed in Section 4.2.3, shaped $(150, 1)$. The probability of specific people flow classes is denoted as $P_i$. The logit value can be calculated as follows:

$$logit = \frac{P_i}{1 - P_i} = \sum_{j=1}^{150} w_j x_j + b \tag{9}$$

where $w_j$ and $x_j$ are the model weight and the input of $j$ th class of the 150 segmentation classes, respectively; and $b$ is the bias of the model. Therefore, the objective function can be expressed as:

$$\underset{w \in R}{argmax} \left\{ \frac{1}{n} \sum_{i=1}^{n} [y_i \log P_i + (1 - y_i) \log(1 - P_i)]^2 - \lambda \|w\|_1 \right\} \tag{10}$$

Following Equation (10), we optimized the SpM to fit the people flow trend ground truth. The weight value graph at the model convergence is shown in Figure 19. The trained SpM weight was shaped as $(150, 10)$, which is the same as that of the Grad-CAM scores. Since SpM was constrained by an L1 regularization, most of the weight values were 0. Therefore, we selected the non-zero values to perform the visualization.

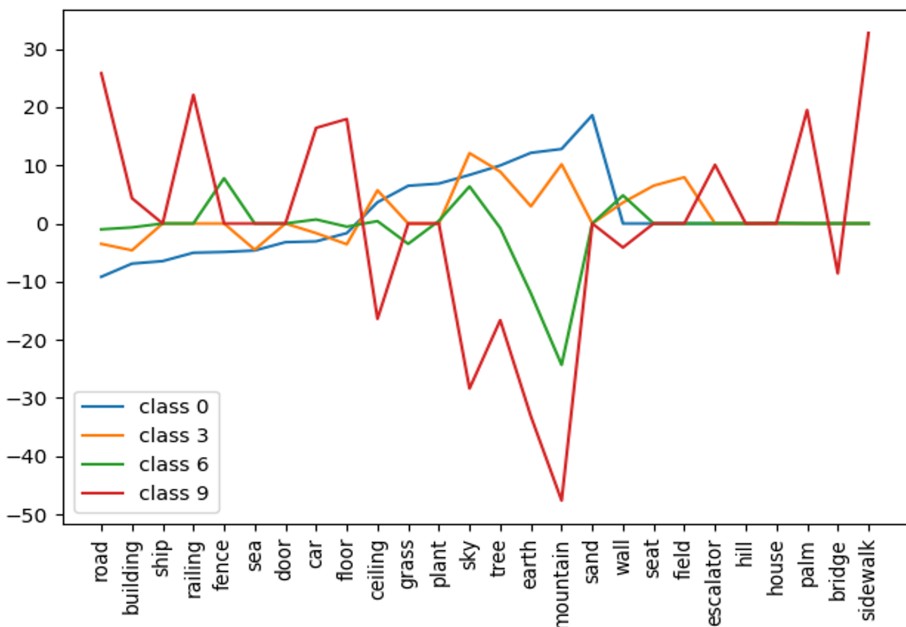

**Figure 19.** SpM weight value visualization.

In addition, as the shapes of the Grad-CAM scores and SpM weight values were both $(150, 10)$, we analyzed this homogeneity trend statistically by filtering out the weights with $pValue < 0.01$ in the *t*-test and compared them with the corresponding Grad-CAM scores.

As the value scale of SpM weights and Grad-CAM scores were different, the horizontal axis means SpM weights and the vertical axis means Grad-CAM scores were normalized. Thereafter, we calculated the correlation coefficient of these two as shown in Equation (11):

$$\rho_{X,Y} = \frac{Cov(X,Y)}{\sigma_X \sigma_Y}, Cov(X,Y) = E[(X - \mu_X)(Y - \mu_Y)] \tag{11}$$

where $\rho_{X,Y}$ is the correlation coefficient; $(X, Y)$ indicate the SpM weights and Grad-CAM scores; $Cov$ is the covariance; $\sigma$ is the standard deviation; $E$ is the expectation; and $\mu$ is the mean value. The result is $\rho_{X,Y} = 0.535$, which proves that the backward objective and subjective explanations were statistically reasonable.

## 6. Discussion

### 6.1. People Flow Trend Estimation

The limitations of this people flow estimation model are as follows: First, the processing speed of the model needs to be considered. In our primary experiment device, Nvidia A100 40G GPU, the inferencing speed can attain maximum 274 frames per second (FPS) and 269 FPS based on Swin-B and ConvNeXt-B, respectively. However, the computation capability of mobile devices is considerably inferior to that of graphics card [58]. Thus, our future work will focus on use of pruning [59], distillation [60], and lightweight backbones [61,62] to improve processing speed and make it more compatible with mobile devices.

In addition, the accuracy of the model on the wide area data and the four detailed patterns is lower than the 78.12% accuracy achieved on the test dataset. This is because the deep learning dataset only considers 3.78% of the wide area image data. The comparison of the accuracy between the test set, wider area data, and specific four patterns shows that the model is not as effective as 78.12% in other parts of the world. To overcome this issue of accuracy loss, we will implement transfer learning [40] and self-supervised [63] approaches to make our workflow more applicable to a global scale.

### 6.2. Ancillary Data Generation and Model Improvement

To improve and explain the people flow trend estimation model, we introduced the human subjective score into the model and used the semantic segmentation result as the pixel categories information to support the explanation.

Regarding the semantic segmentation, we selected UperNet, and the backbone is Swin-B. The model was pretrained on the ADE20K dataset. The semantic segmentation could provide the pixel level classification results, which enables the analysis of the type of pixels exerting the greatest impact on the people flow estimation model. However, the segmentation result could be improved in the future. Considering the annotation cost, we directly deployed the ADE20K pretrained model on our panoramic street view images, and as the features of panoramic images are quite different from the general images, the predictions were not as effective as the original ADE20K dataset. Moreover, considering that the number of the pixel classes defined in the ADE20K dataset was 150, which can cover most of our street view scenes, there are many irrelevant indoor factors that may disturb our further analysis. To address these two problems, redefining appropriate pixel classes to construct the semantic segmentation dataset to perform the fine tuning will be considered in our future work for segmentation.

Regarding the human subjective score extraction, we constructed an image perceptual attribute ranking based on a Siamese network. As the subjective score extractor is a branch of the Siamese network, the total testing accuracy of the Siamese network was 88.23%, which proved that the subjective score extractor was reliable. Thereafter, we concatenated the subjective analysis with the logit value of the people flow estimation model outputs, to improve the performance of the estimation model. An improvement of 78.12% to 78.32%

can be observed in Tables 2 and 3, respectively. This experiment proved that the multi-source data or the explicit expression of image implied features may improve the original end-to-end deep learning model. However, such a complex subjective score extraction method only brought a 0.2% improvement on the original model. This is primarily because we simply concatenated the score with the logit value. Therefore, more reasonable feature fusion methods need to be explored in our future study for the model improvement. In addition, there are many studies that have implemented semantic segmentation to connect the pixel level classification with perceptual attribute more closely [36,49,64]. Therefore, the usage of segmentation and subjective score need to be explored further.

*6.3. Explanation of Deep Learning Processing and Results*

To explain the people flow trend estimation model forward, we quantitatively analyzed the pixel proportion features and subjective score features of each people flow trend class. That is, we already knew the type of scene in which people tended to aggregate; therefore, provided that the pixel distribution and subjective scores are known, the people flow trend can be obtained. However, the quantitative analysis was not sufficiently precise. Thus, analyzing the relationship between the exact change of proportion and score vectors, and the extent of people flow change will be included in our future work with respect to the forward quantitative analysis.

In the backward analysis we analyzed the type of pixels and subjective scores that exert the greatest impact on the people flow trend estimation models. That is, our explanation of the proposed deep learning model could provide large amounts of essential information for applications such as urban planning, disaster management, and transportation planning. The explanations can provide essential information to the concerned professional with respect to the people flow trend in relation to the change of pixels and the subjective scores. Furthermore, this explanation workflow can be generalized to economics and marketing. In our future work, we will consider more fine-grained time slice and movement mode models, to generalize the models to fit for worldwide data, and will deploy our explanation approach to analyze the worldwide relationship commonality between the subjective feeling, streetscape, and people flow trend.

**7. Conclusions**

First, we proposed a deep learning end-to-end people flow trend estimation approach based on panoramic street view images. This approach achieved a maximum total accuracy of 78.12% on the test dataset, and we generalized the model to Kōchi, Japan, spanning across an area of 761.74 km$^2$. The maximum total accuracy was obtained as 72.71% for approximately 1.5 million images. In addition, for further model evaluations, we reprocessed the dataset into four detailed patterns according to the movement and time characteristics of the people living in Kōchi: Day/Stay, Day/Move, Night/Stay, and Night/Move patterns, for which the obtained accuracy values were 64.44, 70.20, 68.66, and 69.96%, respectively.

Second, we selected the UperNet semantic segmentation method with Swin-B backbone and directly deployed the model that was pretrained on the ADE20K data to provide pixel categories information for further analysis. Thereafter, we improved the human subjective score extraction approach based on a Siamese network to provide subjective scores for further deep learning explanations. Furthermore, the subjective scores can also be used to improve the performance of the people flow estimation model. The total accuracy was improved to 78.32%, and the detailed patterns were improved to 64.62, 70.40, 68.80, and 70.12%.

Finally, we explained the people flow trend estimation forward results. We quantitatively analyzed the pixel proportion features and subjective score features of each people flow trend class. Thereafter, we implemented the Grad-CAM deep learning visualization method and proposed to perform the quantitative analysis on the Grad-CAM score. Furthermore, with respect to the deep learning visualization method, we proposed a gradient impact method for subjective factors analysis. Thus, we explained the people flow esti-

mation model backward by analyzing the type of pixels and subjective scores that exert the greatest impact on the deep learning model. Lastly, to prove the plausibility of this backward analysis, we compared this backward explanation with a more explainable SpM regression method and obtained a correlation coefficient of 0.535.

This study intuitively analyzed the relationship between the streetscape, the subjective feeling with respect to the street view images and the people flow trend. Directly connecting the subjective feeling and streetscape with the people flow trend with deep learning approach, and the deep learning explanation method for the relationship analysis, provided a novel consideration for urban planning, disaster management, traffic planning, and marketing fields, which is more promising and intuitive than the use of indices such as walkability [65], pedestrian [66], and moveability [67] indices. To further explore the efficiency and prospect of this study, the following aspects will be considered for our future work.

First, we need to accelerate the inference speed of our model by pruning, distillation, and considering a lightweight backbone, and by deploying the transfer learning and self-supervised approaches to improve the generalization ability of the model. Second, we need to redefine the segmentation classes and build our dataset to fine-tune the segmentation model. Furthermore, we need to explore more efficient feature fusion methods and try to connect the semantic segmentation with the subjective score with improved closeness. Finally, we will try to generalize our workflow for applications in the fields of economics by constructing a deep learning economic index prediction model to provide information for decision makers to implement changes considering the key factors.

**Author Contributions:** Design and conceptualization, C.Z. and Y.O.; methodology, C.Z. and S.C.; research and experiments, C.Z. and S.C.; writing—original draft preparation, C.Z.; writing—review and editing, T.O. and Y.O.; supervision, Y.O., T.O. and Y.S. All authors have read and agreed to the published version of the manuscript.

**Funding:** This work was supported by JSPS KAKENHI Grant Number 20K15001 and 22K04490.

**Data Availability Statement:** Not applicable.

**Conflicts of Interest:** The authors declare no conflict of interest.

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
