# Peer review of "People Flow Trend Estimation Approach and Quantitative Explanation Based on the Scene Level Deep Learning of Street View Images"

_remotesensing, doi:10.3390/rs15051362_

Round 1

Reviewer 1 Report

1. In Section 4.2.1, it is necessary to explain how to construct pairs between photos for subjective score calculation. (e.g. criteria for selection)

    That is, the preference result may be different depending on which pair is formed.

2. 2. Figure 6, some words also show semantic overlap or similarity as follows.

   -Comfortable vs Neat vs cozy

   -Like vs Interesting

   - lively vs Bright

   - Attractive as Like

  In other words, repeated surveys with similar expressions may cause data duplication or bias, which will affect the subjective score. Thus, the authors shall explain or analyze this issue.

3. The authors shall explain how the words (open, friendly, etc) in the questionnaire are related to the flow trend. From a traveler's point of view, there seems to be a connection, but from a business perspective, the connection seems small. (Line 56 of the manuscript)

4.The authors collected GPS information (Konzatsu Analysis) of random people for 'People Flow Data'. Was the subjet score survey also conducted through these people? 

If the people in the GPS and the people in the survey are different, does the accuracy of flow trend estimation change?

5. Section 4.1.2 (General information about deep learning) is appropriate to move to Introduciton or related works section.

6. The introduction of existing technologies in Section 4.2.1 should also be moved to related work, and explanations shall focus on the main ideas of this manuscript.

7. The first paragraph of Section 6.1 is a reiteration that has already been mentioned several times in the manuscript. Any redundant explanations throughout the entire manuscript shall be minimized.

8. It would be good if the accuracy calculation method (formula) was included.

Author Response

Thank you very much for your comments, especially for pointing out the problems of subjective perception score estimation, we will have a point-to-point response to your comments and concerns. Please see the attachment.

Reviewer 2 Report

The new scene level end-to-end deep learning approach, using a combination of street view images and the human subjective score of each street view, to estimate the people flow trend is proposed in this manuscript.

In a formal sense, the manuscript is well written because it contains all the necessary elements. Unfortunately, the work lacks theoretical evidence for the claims made, so from the point of view of scientific contribution, it is a modest achievement. The manuscript lacks target, noise, and measurement theoretical models that should be listed in Chapter 2, right after the Introduction. Also, a mathematical model of the proposed algorithm should be given, so that the reader can check the effectiveness of the proposed method independently of the author, on his own platform (for example in Chapter 3).

It is desirable to compare the proposed method in terms of efficiency in the part of the simulation results with similar methods that solve the same problem.

Suggestions for improvement the manuscript:

  • State the theoretical models of target, measurement and noise.
  • In Chapter 3, state the mathematical model of proposed algorithm.
  • Compare the proposed method with similar methods that solve the same problem

Author Response

Thank you very much for pointing out the lack of theoretical/mathematical model in this study, we will have a point-to-point response to your comments and concerns. And please see the attachment.

Reviewer 3 Report

(1)This paper proposes a method to predict the population size for four situations, day/stay, day/move, night/stay, night/move. The title 'People Flow Trend Estimation ***' may be unsuitable. This study only predicts the numbers of the population under stay or move status in each grid. However, people flow emphasizes the movement of people between different locations

(2)People flow trend data used to train the model are from "Konzatsu-Tokei (R)" GPS data; what is the coverage rate of this mobile phone service provider?

(3)From street view images, we can only see people on the street; we cannot see people indoors. Mobile phone GPS data cannot distinguish between the street and indoor population. Does this difference have a potential influence on your results?

(4)There are some grammar mistakes in the paper; the manuscript should be thoroughly edited for grammar.

Author Response

Thank you very much for your comments, especially for pointing out the different concepts of ‘population’ and ‘people flow’, we will have a point-to-point response to your comments and concerns. And please see the attachment.

Round 2

Reviewer 1 Report

The authors highly improved the qualtity of the manuscript by reflecting reivewer's comments.

Reviewer 2 Report

The new scene level end-to-end deep learning approach, using a combination of street view images and the human subjective score of each street view, to estimate the people flow trend is proposed in this manuscript.

The authors have conducted a large number of experiments in real conditions and with images taken in real scenes, and the manuscript has its own value. Unfortunately, from the point of view of the academic public, the manuscript does not have the necessary scientific contribution, due to the lack of theoretical evidence for the claims made in the manuscript.

The changes made in the revised manuscript are not enough to improve the manuscript, so that it could be published in the journal